# Impact of military service on physical health later in life: a qualitative study of geriatric UK veterans and non-veterans

Victoria Williamson,® Hannah Harwood, Karla Greenberg, Sharon A M Stevelink, N Greenberg

King's College London, London, UK

**Correspondence to**
Dr Victoria Williamson;
victoria.williamson@kcl.ac.uk

## ABSTRACT

**Objectives** Military veterans often experience physical health problems in later life; however, it remains unclear whether these problems are due to military service or are a feature of the ageing process. This study aimed to explore veteran and non-veteran perceptions of the impact of their occupation on their physical well-being later in life.
**Design** Semi-structured qualitative interviews analysed using thematic analysis.
**Setting** Interviews were conducted face-to-face in participants' homes or via telephone.
**Participants** 35 veterans (≥65 years), 25 non-veterans (≥65 years) were recruited, as well as a close companion of all participants for triangulation (n=60).
**Results** Most veterans reported good physical health later in life which they attributed to the fitness they developed during military service. However, several veterans described challenges in maintaining their desired level of physically activity due to new commitments and limited sports facilities when they left service. Fewer non-veterans had experienced work-related fitness activities or exercise in their civilian jobs. Ongoing physical health difficulties, such as deafness, were perceived to be due to exposure to workplace hazards and appeared more common in veterans compared with non-veterans. Veterans also described greater reluctance than non-veterans to seek medical treatment for physical health difficulties, which could be challenging for close companions who had to provide informal care.
**Conclusions** Military service was largely perceived to be beneficial for physical well-being; although when occupation-related physical health problems were experienced, many veterans were unwilling to seek treatment. These findings may inform clinicians of the needs of older veterans and highlight potential barriers to care.

## Strengths and limitations of this study

► Large-scale qualitative study (n=120) which recruited veterans and non-veterans aged over 65 years and a close companion of each participant.
► Semi-structured, qualitative interviews were conducted face-to-face in participant's homes or via telephone.
► Inclusion of the views of a close companion (spouse, child, close friend) to triangulate the study findings.

However, as over 65's in the general population also commonly suffer from various physical health problems,[5 6] whether health issues experienced by older veterans are related to UK military service or are a feature of ageing remains unclear.

The limited available evidence indicates that UK AF veterans may be more likely than the general population to experience physical health conditions that affect their daily functioning. For example, working age UK AF veterans have higher rates of hearing and musculoskeletal problems compared with non-veterans.[2] Nonetheless, military personnel are often fitter and healthier at the time of enlistment than the general population, leading to a phenomenon termed the 'healthy warrior effect'.[7] This may explain why a 25% lower mortality risk for older Australian veterans compared with non-veterans has been observed.[7] Moreover, the high-levels of physical activity required in the AF as part of training, sport or deployment activities may offer some protective effects for health conditions later in life, with midlife exercise found to significantly reduce the risk of later high cholesterol and dementia (eg,[8]). Finally, physical health problems can often have an impact on family members and caregivers. How health problems in working-age individuals can affect their families has been examined in previous studies,[9 10] however

The UK Armed Forces (AF) veteran population is becoming increasingly elderly, with approximately 64% of UK AF veterans over the age of 65.[1 2] Despite the sizeable and growing elderly UK AF veteran community, relatively little is understood about the impact of military service on the well-being of older UK AF veterans. Limited literature, mostly from the USA, suggests there may be several health implications from military service.[2–4]

comparatively little is known about the experiences of caregivers of older ex-military personnel.

A deeper understanding of the perceived long-term impact of AF service on physical health may inform clinical practice, identify whether there is a need to alter the way in which services for this population are provided and highlight areas for occupational policy change. We conducted in-depth, qualitative interviews with older veterans, older non-veterans and a close companion (ie, spouse, child, close friend) of each veteran/non-veteran participant. We aimed to explore perceptions of the impact of military versus non-military occupation on physical well-being later in life.

## METHOD
### Study design
This was a qualitative study using in-depth interviews. The qualitative approach utilised allowed for the exploration of issues that were most salient to participants and the subjective meaning attached to their career and well-being. The present study was nested within a larger programme of research[11] examining the impact of military service on physical, psychological and social functioning later in life.

### Participants
Four subsamples of participants were recruited: (1) 25 older veterans (≥65 years, five or more years of regular AF service) not known to have mental health problems, (2) 25 older non-veterans not known to have mental health problems (≥65 years, five or more years of work in a non-military occupation), (3) 10 older veterans with self-reported experiences of mental health issues (≥65 years, five or more years of AF service) and (4) a close companion of all participants (≥18 years, n=60). The 25 veterans without mental health diagnoses and 25 non-veterans were age (+/-5 years) and gender matched. Exclusion criteria included: brain damage or cognitive impairment, inability to speak English or current self-injurious behaviour/suicidal intent.

## ASSESSMENTS
### Physical health
Veteran and non-veteran physical health and quality of life were measured using the 12-item Short Form Health Survey (SF-12)[12] and the EuroQol (EQ-5D).[13] Summary scores for the physical health component of the SF-12 were used (maximum possible score of 100), with high scores indicating better health state. The visual analogue scale (VAS) of EQ-5D was used which asks respondents to rate their overall health with 'worst imaginable health state' set at 0 and 'best imaginable health state' set at 100.[13] The reliability and validity of both measures have been evidenced. These questionnaire assessments were completed by post (n=62) or online (n=58).

## Qualitative interview schedule and procedure
The semi-structured interview questions focused on participants perceptions of their career, the impact of their career on their physical health and their experiences of informal care for physical health problems (see online supplementary file 1). Close companions were asked for their perceptions of the veteran or non-veteran's career and its impact on physical health, as well as their own experiences of caring for the veteran/non-veteran.

The majority of interviews were conducted face-to-face (n=82), with 38 conducted via telephone. All participants gave written (if interview conducted in person) or audio-recorded verbal (if interview conducted via telephone) informed consent for their participation. Interviews were conducted by study researchers who had training and experience in qualitative methods. Interviews lasted for an average of 67 min (median=65 min, SD=0.02, IQR=52:29 to 01:20:56 min). All interviews were digitally recorded and transcribed verbatim. Interviews were completed by 120 participants to represent a wide range in occupations and ages to reflect the possible diversity of opinions. Thematic saturation was achieved, and this was determined by the research team when no additional themes were found from the reviewing of successive data.[14]

## PROCEDURE
Purposive sampling was used. As previous research has highlighted the relationship between poor mental health and physical health problems,[2 15] the aim of the purposive sampling of veterans with self-reported mental health problems was to allow for further exploration of potential links between occupation and health concerns. Veteran and non-veteran participants were initially identified by the clinical care team following attendance at National Health Service general practitioner surgeries, walk in centres, community groups or mental health services. The clinical care team identified potentially eligible participants using study inclusion/exclusion criteria. The clinical care team sought permission for patient contact details to be shared with the research team. Given this agreement, participants were contacted by researchers with additional study information. Participants could also self-refer to the study via study advertisements. Companions were recruited by asking veteran/non-veterans if they had a spouse, child or close friend willing to participate with them. Veteran/non-veterans provided their companion's contact details and companions were then contacted by the research team and sent a study information pack.

A total of 86 veteran/non-veteran participants were approached to take part in the study, 60 of which were recruited (69.8% recruitment rate). Eligible individuals who did not participate were largely not contactable or had no close companion to take part. All participants

were recruited between March and December 2017. All participants were given a £20 voucher as a thank you for their time.

## DATA ANALYSIS
### Quantitative analysis
Data were entered and analysed using the statistical analysis software package SPSS (V.24). Missing data on the EQ-5D VAS and SF-12 were excluded from the analysis (EQ-5D VAS data missing for three non-veterans and one veteran, SF-12 data missing for five non-veterans and six veterans). T-tests were used to determine whether statistically significant differences between groups existed, with p values <0.05 used to indicate statistical significance.

### Qualitative analysis
All transcripts were entered into NVivo V.11 (QSR International). Data were analysed using thematic analysis using the steps described by Braun and Clarke[16]: reading and re-reading the data, producing codes, searching for and developing early themes and revising and refining themes. Thematic analysis was utilised as it is a method suitable for larger sample sizes and analytical strategy used to identify patterns of meaning across the data set as a whole, in keeping with the study's objective of exploring the impact of occupation across the life-course. An inductive analytical approach was used with initial codes and themes proposed by VW. Data collection and analysis took place simultaneously to allow emerging topics of interest to be investigated further in later interviews and to determine whether thematic saturation had been reached.[14] A reflexive journal was kept throughout data collection and analysis by the primary researcher (VW) to recognise the influence of the researcher's prior experiences, thoughts and assumptions and avoid premature or biased interpretations of the data.[17] To ensure reliability, all codes and themes were independently reviewed by authors VW and HH. Disagreements between authors were infrequent and were resolved following discussion and re-examination of the data. Peer debriefing was conducted to further enhance credibility, where discussions were held about the emerging findings and feedback regarding data interpretation and analysis was sought from co-authors SAMS, KG, and NG.

## PATIENT INVOLVEMENT
The study involved patients in shaping the study design and outcome measures as the study materials were piloted with veterans (n=3) and non-veterans (n=2). Patients were also involved in the study recruitment as recommendations for potential recruitment sites were sought from patient participation group meetings. The findings were disseminated to participants via a study newsletter.

## RESULTS
### Descriptive information
Participant demographic information can be found in table 1. Overall, the 35 veterans had between 5 to 40 years of AF service, with 34.3% serving in the Naval Services, 51.4% serving in the Army and 14.3% in the Royal Air Force (see table 1). Close companions (n=60) were largely spouses (n=46, 76.7%) or close friends (n=8, 13.3%) and had a mean age of 68.4 years (SD 10.6).

Differences in scores on the SF-12 and EQ-5D were not statistically significant between veteran, veterans with mental health diagnoses and non-veteran samples (p>0.05, see table 1).

## QUALITATIVE RESULTS
Three key themes emerged from the data regarding experiences and perceived impact of occupation-related exercise, exposure to workplace hazards and the effects on health and perceptions of and responses to physical health problems. Anonymised participant comments are provided to illustrate the findings in table 2, and all participants have been given a pseudonym.

### Occupation and physical exercise
The majority of veterans reported high-levels of physical exercise in the AF, which was not described by non-veterans in civilian roles. Many veterans thought they had been very physically fit due to the military training exercises and sports. This high volume of exercise during their military career was seen as a key reason for veteran good physical health later in life. Many veterans continued to exercise once leaving the AF and keeping physically active was a central part of daily life in retirement. In roles where non-veterans had the opportunity to do some exercise (eg, manual labour), this activity was considered to have kept them in good health at the time but was often seen as having no long-term benefits.

Nonetheless, several veterans stated they had chronic arthritis and other musculoskeletal health complaints in older age which were thought to be due to the physically demanding activities required in the AF (eg, parachuting, marching exercises, etc). These health problems were less commonly described by non-veterans and were not thought to be related to occupation.

On leaving the AF, a minority of veterans experienced difficulties continuing physical activity, either due to the long working hours of their civilian job, being employed in a more sedentary civilian role (eg, heavy goods vehicle driving, security, etc), limited access to sports facilities or due to a military-related physical injury. Veteran difficulties continuing exercise were also discussed by their close companions.

### Impact of occupational health and safety
Participants in both veteran and non-veteran samples reported sustaining physical injuries in the workplace, with the majority of injuries viewed as being due to a lack of protective equipment or poor health and safety regulations (eg, lack of ear defenders, limited protection available against hazardous substances).

**Table 1**  Participant demographic information

| Demographic | Veterans with MH diagnosis n (%), n=10 | Veterans n (%), n=25 | Non-veterans n (%), n=25 | Close companions n (%), n=60 |
|---|---|---|---|---|
| | n=10 | n=25 | n=25 | n=60 |
| Age M (SD) | 71.8 (6.5) | 74.6 (6.9) | 75.3 (7.5) | 68.4 (10.6) |
| Gender | | | | |
| Male | 7 (70.0) | 22 (88.0) | 22 (88.0) | 9 (15.0) |
| Female | 3 (30.0) | 3 (12.0) | 3 (12.0) | 51 (85.0) |
| Ethnicity | | | | |
| White British | 10 (100.0) | 25 (100.0) | 24 (96.0) | 59 (98.0)* |
| Asian/Asian British | n/a | n/a | 1 (4.0) | |
| Service branch | | | | |
| Naval services | 5 (50.0) | 7 (28.0) | n/a | n/a |
| Army | 4 (40.0) | 14 (56.0) | | |
| RAF | 1 (10.0) | 4 (16.0) | | |
| Service length M (SD) | 20.5 (12.8) | 19.6 (8.4) | n/a | n/a |
| Non-veteran professions | | | | |
| Doctor/nurse | n/a | n/a | 3 (12.0) | n/a |
| Managerial/office | | | 6 (24.0) | |
| Manual labour | | | 5 (20.0) | |
| Small business owner | | | 3 (12.0) | |
| Police | | | 1 (4.0) | |
| Engineer/scientist | | | 4 (16.0) | |
| Civil service | | | 3 (12.0) | |
| CC relationship | | | | |
| Spouse | 7 (70.0) | 16 (64.0) | 23 (92.0) | n/a |
| Child | 1 (10.0) | 3 (12.0) | 1 (4.0) | |
| Close friend | 2 (20.0) | 5 (20.0) | 1 (4.0) | |
| Niece/nephew | n/a | 1 (4.0) | n/a | |
| Physical health score M (SD) | 53.4 (2.4) | 54.0 (2.9) | 53.6 (2.3) | n/a |
| Quality of life M (SD) | 66.4 (16.4) | 77.2 (18.1) | 70.7 (21.2) | n/a |

Data missing for three non-veterans and one veteran (no mental health diagnosis) on the EQ-5D VAS. Data missing for five non-veterans and six veterans (no mental health diagnoses) on the SF-12.

*This demographic information was missing for one participant. No significant differences between SF-12 scores were found between veterans and veterans with MH diagnoses (t(33)=0.96; p=0.35), veterans without mental disorders and non-veterans (t(48)=-1.20; p=0.24) and non-veterans and veterans with mental health diagnoses (t(33)=0.11; p=0.916). No significant differences between EQ-5D VAS scores were found between veterans and veterans with MH diagnoses (t(32)=1.619; p=0.12), veterans without mental disorders and non-veterans (t(44)=-1.11; p=0.27) and non-veterans and veterans with mental health diagnoses (t(30)=0.57; p=0.57).

CC, close companion; CC relationship, the close companion's relationship to the veteran or non-veteran; EQ-5D, EuroQol; M, mean; physical health score, mean score on SF-12 physical health component; quality of life, mean score on the EQ-5D VAS; RAF, Royal Air Force; SD, standard deviation; service length, number of years in military service; SF-12, Short Form Health Survey; VAS, visual analogue scale; veterans with MH diagnoses, veterans with a previous self-reported diagnosis of mental health disorders.

Veterans more frequently reported long-term health difficulties due to a lack of protective equipment compared with non-veterans. Veterans often described experiencing hearing loss due to a lack of ear defenders in loud engine rooms or on shooting ranges, as well as skin cancers believed to be caused by having no sun protection during training activities or deployment.

For non-veterans, those in physically demanding jobs often reported strain injuries due to heavy lifting or spending several hours standing. Physically demanding jobs were thought to exacerbate existing physical health conditions (eg, existing musculoskeletal problems made worse by heavy lifting). Some non-veterans in healthcare roles or jobs requiring foreign travel (eg, foreign office,

**Table 2**  Themes and subthemes following thematic analysis

| Themes and subthemes | Findings | Verbatim quotes |
|---|---|---|
| **Occupation and physical exercise** | | |
| High volume of exercise in AF | Veterans reported high-levels of physical exercise during their military career and this was considered a key reason for their good physical health later in life. Few opportunities for exercise were described in non-veterans in civilian roles. | Veteran 1: I've always been pretty healthy, I used to do a lot of sport when I was in the Army…. (Now) I like to go for walks to keep me fit and I (have) a treadmill… So, mentally and physically I'm not too bad for my age. |
| AF exercise and health problems | The high volume of exercise during military service was thought to have caused chronic health problems in some veterans later in life (eg, arthritis). These problems were less common in non-veterans and reportedly unrelated to occupation. | Veteran 2: It had an effect on my knees, osteoarthritis… When I joined the Army I did an awful lot of athletics… The Army doesn't do you any good. I mean you've only got to come to a parade here and see the old Regimental Sergeant Major banging his feet on the ground and all that sort of thing. And so, my knees are shot. |
| Difficulties continuing exercise on transition from the AF | Veterans described difficulties in continuing to exercise on transitioning from the military for several reasons, including a lack of sports facilities and more sedentary civilian roles. | Veteran 3: No, I haven't done any real sport since I went outside. There hasn't been the opportunity for it really…The (sports) centre here, I need two buses to get to it so… you know, puts you off a bit. |
| **Impact of occupational health and safety** | | |
| Workplace injuries/illnesses | Both veterans and non-veterans described experiencing workplace injuries/illnesses which were often due to poor health and safety regulations. In particular, hearing loss was commonly described by veterans, whereas strain injuries were frequent discussed by non-veterans. | Non-veteran close companion 1: Through his plodding around the streets and that…his feet are very bad…They're deformed… He's had waterworks problems, he's got arthritis in both of his shoulders which we think is due to him being out in all weathers policing… he'd be out overnight in all weathers and get drenched and we think that's not helped his arthritis. |
| Impact of occupational hazards on the family | The health of family members was thought be affected by veteran and non-veteran's exposure to some workplace hazards. Indirect hazard exposure was believed to cause serious illness in family members which often caused veterans/non-veterans to feel guilty and concerned for their well-being. | Non-veteran 3: I was on duty… and the nurse in charge…said 'I've heard you're pregnant, great! Don't come any closer, my little boy's got rubella!' And I didn't… I didn't even go into the room, and 2 weeks later, I had rubella…and that's why my (child) was profoundly deaf. |
| **Perceptions of and responses to health problems** | | |
| Reluctance to access care | Reluctance to seek medical treatment for physical health problems both during and following military service was common in veterans. This reluctance was largely not found in non-veterans. Unwillingness to access care was reportedly due to an AF instilled need to feel and be seen as 'tough.' | Veteran 8: It's a man thing! Compounded by the Army. You can't be seen to be a wimp! You are a wimp, but you can't be seen… You've got to keep a stiff upper lip… if you go to anything like an injection, there's no way you flinch! It might hurt, but you don't flinch! No! |

**Table 2** Continued

| Themes and subthemes | Findings | Verbatim quotes |
|---|---|---|
| Experiences of care and support | Veteran close companions described that ill/injured veterans were extremely reluctant to accept (in)formal care, often due to embarrassment or pride. Unwillingness to accept care was less common in non-veterans. | Veteran close companion 4: He did not enjoy having things done for him, where he couldn't reach his feet or his back to wash properly…So I had to do that for him and he didn't like that. Got very, very upset over that…He thought I was trying to take control of his life. Which I wasn't!… (And) we would have an argument. |
| Impact of providing care on the family | Providing care to unwilling veterans could be distressing and frustrating for close companions, who often felt their efforts were not appreciated. Close companions of non-veterans described facing less resistance to their provision of support. | Veteran close companion 3: (The doctors) found this cancer on him. And then he said 'I'm not having any treatment'… I said 'Look, there's more than you here…You've got (your) grandson…and all (your) friends… You've just got to be a bit considerate'. Anyhow, he had (the treatment). |

AF, Armed Forces. All participants have been assigned a pseudonym.

employment with an international company, etc) also described that they had contracted serious illnesses (ie, shingles, dysentery, etc). Such injuries/illnesses were often felt to have long-lasting effects, with physical pain or disability experienced into retirement.

### Impact of occupational hazards on the family

It reportedly became more challenging for close companions to cope with veteran and non-veteran's occupation-related health problems later in life, particularly in cases of veteran/non-veteran hearing loss or debilitating illness. These close companions described feeling lonely as well as frustrated that their plans to exercise or travel as a family in retirement were now not feasible.

The health of close companions or other family members could also reportedly be affected by veteran and non-veteran's exposure to workplace hazards. For example, some close companions discussed being exposed to asbestos on veteran/non-veteran's work clothes. This indirect exposure was thought to cause serious illness, and family members reportedly experienced cancer, infertility, breathing problems and other conditions as a result. When family members became unwell in such circumstances, veterans and non-veterans disclosed experiencing significant guilt and anxiety for their family members' well-being, often taking considerable steps to address the perceived harm they had caused (eg, setting up a charity to help others with similar health problems, becoming their full-time carer, etc).

### Perceptions of and responses to health problems

Veterans with and without diagnoses of mental health problems described experiencing similar self-reported physical health problems (eg, arthritis, diabetes, high blood pressure, hearing loss, etc). Veterans without diagnoses of mental health difficulties more often considered that their physical health conditions related to AF service compared with veterans with mental health diagnoses. Few other differences in terms of physical health problems were described between the two subsamples.

Reluctance to seek medical treatment for physical health problems both during and following military service was commonly reported in veterans. This reluctance was largely not described by non-veterans. Unwillingness to admit physical health problems and access treatment was reportedly due to a sense of self-sufficiency instilled by the AF and a need to feel and be seen as 'tough'.

### Experiences of care and support

Close companions detailed how they play a central role in encouraging veterans to access medical treatment, often accompanying them to appointments and ensuring medication was taken. In cases of serious illness/disability, many close companions of veterans described that veterans were extremely reluctant to accept informal care, such as assistance with getting dressed, daily hygiene, etc, often due to embarrassment or pride, and unwillingness to accept care was less common in non-veterans.

Providing support and care to unwilling veterans could be a distressing and frustrating experience for close companions who often felt their efforts were not appreciated. To cope, close companions positively reframed their experience, for example describing that the veteran would provide them with the same care if needed or seeking support from other family members and friends. Close companions of non-veterans described facing less resistance to their provision of support.

 Williamson V, et al. BMJ Open 2019;9:e028189. doi:10.1136/bmjopen-2018-028189

## DISCUSSION

The aim of this study was to explore the perceived impact of military and non-military occupations on physical health later in life. Our findings illustrate the perceived considerable benefits of occupation on well-being, particularly for those who served in the AF, such as long-term good physical health as a result of physical exercise. At the same time, these results also highlight the significant long-term implications of workplace practices on well-being, including chronic physical health problems due to a lack of protective equipment and beliefs regarding the wider impact of such workplace hazards on an employee's family.

Our findings delineate the positive influences of occupation on health, including the perceived long-lasting benefits of physical exercise in the AF. Physical exercise is associated with lower risk of health problems later in life, including dementia[8] and cardiovascular issues.[18] This may potentially mean AF veterans are at an advantage as opportunities for occupational physical activity were more limited in non-veterans and perceived long-lasting health benefits of occupational exertion in this group were not described. However, as we found no significant differences in physical functioning between veterans and non-veterans on the SF-12 and EQ-5D VAS, this finding must interpreted cautiously. Moreover, some difficulties continuing physical exercise on leaving the AF were experienced due to logistical barriers or musculoskeletal injuries. This is in line with previous research in working age USA and UK veterans[15 19 20] and provides insight into why some veterans may be vulnerable to obesity on leaving the AF.[21 22] Interventions to increase physical activity in non-veteran workplaces and additional support for veterans to facilitate exercise on leaving the AF may be beneficial. Nonetheless, it is important to stress that this exercise must be well managed, as veterans also reported experiencing musculoskeletal health complaints which they believed were caused by very strenuous AF physical activity.

Previous research into the impact of military service on physical health has also highlighted that, compared with the general population, working-age veterans are more likely to experience hearing difficulties, musculoskeletal problems and arthritis on leaving service.[2 23 24] The findings of this study provide support for these results, offer insight into why such physical health problems potentially occur, for example that hearing loss occurred due to lack of ear defenders, and what the perceived long-term implications can be. As ongoing health problems linked to lack of protective equipment were found in both veteran and non-veteran groups, this may reflect occupational health and safety standards of the time (eg, 1950 to 1990's[25 26]). In veterans, it could also reflect the inability of the Ministry of Defence to apply health and safety legislation in an unpredictable conflict zone.[27] Given the adverse impact of workplace hazards, this highlights the importance of effective health and safety procedures (eg, regular safety inspections, employee training, etc) and the continued need for the employers to ensure appropriate measures are in place to protect personnel. Participants believed that their family members could also be indirectly exposed to workplace hazards which were thought to contribute to physical health problems (eg, respiratory difficulties, cancer, etc). While ascertaining whether family members' health problems were a result of veteran/non-veteran occupational practices or other factors, such as attribution bias, was beyond the scope of this study, these findings are not inconsistent with previous research (eg,[28]), and suggest a need for continued research regarding the long-term health implications of hazardous work environments. These results also suggest that employers should provide employees family members with accurate information about potential health threats in order to safeguard against potential ill-effects, allay fears and inform them when and how to access medical support.

Another key theme emerged in relation to treatment seeking as veterans appeared to be more reluctant than non-veterans to seek medical treatment for physical health problems or accept informal care - reportedly due to embarrassment or an AF instilled need to be 'tough'. As delays to necessary medical care can adversely impact well-being,[29] this behaviour is concerning. As efforts have been successfully made to encourage help-seeking in those personnel/veterans experiencing mental health difficulties (eg,[30]), a similar approach for promoting treatment seeking for physical health issues may also be beneficial. Furthermore, close companions often provided support for veterans to access medical treatment as well as informal care. In line with previous research in carers, refusal of care could be a challenging and distressing experience for some.[31 32] Older caregivers who provide care to a reluctant spouse have been found to report long-term marital unhappiness, feelings of entrapment in the role of carer and distress from witnessing the deterioration in their spouse's condition.[33] Therefore, the families of older veterans may benefit from additional support and guidance related to their role as a caregiver. Further research is needed to ensure the provision of such support is accessible and appropriate.

The current study provided a perspective on an important, yet understudied, group in examining the impact of military service on well-being in older UK AF veterans. We recruited a diverse sample of veterans and non-veterans from multiple professions and took steps to ensure the data was robust by triangulation with close companions. At the same time, there are limitations that must be considered. First, this study largely included male veterans/non-veterans, although this is broadly consistent with the gender distribution of the UK AF. Second, this study included only those veterans and non-veterans who agreed to contact by the research team or chose to self-refer in response to study advertisements. This may mean that participants had particularly salient occupation-related issues they wished to discuss. Third, while the key outcome measure of the study was the qualitative interview, a small proportion of participants did not fully

...

complete questionnaire assessments resulting in missing data which must be taken into consideration. Furthermore, the inclusion of non-veterans with diagnoses of mental health problems in future studies would further our understanding of the potential relationship between occupation, physical and mental health. Finally, given the qualitative nature of the study, the findings reflect the lived experiences of a sample of veterans, non-veterans and close companions and large-scale quantitative investigations would be useful in determining the generalisability of the results.

Despite these limitations, this study adds to the very limited research into the long-term physical health implications of military service on older UK veterans in several ways. First, by directly comparing the experiences of veterans and non-veterans, our results suggest that military service may have considerable beneficial effects on physical health later in life due to greater opportunities for exercise. Second, these findings demonstrate that poor health and safety practices, particularly in the AF, may lead to physical health problems which can impact long-term functioning. Finally, this research highlights that additional advice and support may be beneficial to veterans and their families to facilitate veteran access to medical treatment which could ultimately improve their well-being and reduce familial stress.

**Acknowledgements** The authors would like to thank Solent NHS Trust and its PPG for the guidance and support they provided throughout this project.

**Contributors** VW, SAMS, HH, KG, NG made substantial contributions to the conception of the study and analysis and interpretation of data for the study; drafted the article and revised it critically for important intellectual content; gave final approval of the version to be published and agreed to be accountable for all aspects of the article in ensuring that questions related to the accuracy or integrity of any part of the article are appropriately investigated and resolved. VW and HH contributed to acquisition of the data.

**Funding** This research was supported by a grant from the Royal British Legion's Aged Veteran's Fund, grant number AVF-TRBL03 (NG). This paper represents independent research part-funded by the National Institute for Health Research (NIHR) Biomedical Research Centre at South London and Maudsley NHS Foundation Trust and King's College London (SS). The views expressed are those of the authors and not necessarily those of the NHS, the NIHR or the Department of Health and Social Care.

**Competing interests** None declared.

**Patient consent for publication** Not required.

**Ethics approval** Ethical approval for this study was granted by the NHS Camden & King's Cross Research EthicsCommittee (17/LO/0077).

**Provenance and peer review** Not commissioned; externally peer reviewed.

**Data sharing statement** No additional data are available.

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
