## [Reviewer comments · BMJ Open]

ARTICLE DETAILS

TITLE (PROVISIONAL)	The impact of military service on physical health later in life: A qualitative study of geriatric UK veterans and non-veterans.
AUTHORS	Williamson, Victoria; Harwood, Hannah; Greenberg, Karla; Stevelink, Sharon; Greenberg, N

VERSION 1 – REVIEW

REVIEWER	Bonnie M. Vest Research Assistant Professor, University at Buffalo, USA
REVIEW RETURNED	11-Jan-2019

GENERAL COMMENTS	Review of: ID bmjopen-2018-028189 "The impact of military service on physical health later in life: A qualitative study of geriatric UK veterans and non-veterans." This manuscript presents the results of a qualitative study examining the impacts of military service on physical health among older veterans. The study has an impressive sample size for a qualitative work and gathered information from both comparative (non-veteran) and corroborative samples (caregivers). However, while the research question is an interesting and important one and the paper has potential, there are some questions, particularly around the methods and the manner of presenting the results, which need to be addressed. METHODS:  • First, the authors need to be sure they are following the COREQ guidelines for reporting on qualitative research. In particular, justification for why a qualitative approach was chosen for answering this question and how the sample size was determined need to be provided. • Likewise, justification for the groups chosen for the interviews needs to be provided. Why were veterans separated into those with and without mental health diagnoses? This does not seem to be central for the research question being addressed, and should be explained further. • On page 7 in the interview procedure section, the authors state that thematic saturation was achieved. How was this determined? • Providing more context of the larger study would be helpful. It is unclear from the procedures (pg 7) what type of clinical context participants were recruited from, and how this may have influenced the study sample. What criteria did clinic staff use to identify potential participants for the purposive sample? • Page 8, Qualitative Analysis: More detail is needed in regards to the analysis process—how were themes identified? How many individuals participated in the analysis process? What was the method for the peer debriefing? More detail is needed.
--

	RESULTS:  • The organization of the results section makes it difficult to follow. While the authors repeatedly emphasize comparisons between veterans and non-veterans, the quotations provided in the text are often drawn from only one of these groups, which makes it difficult for the reader to evaluate the differences. The strength of qualitative methods is the ability to demonstrate the nuance and complexity of a phenomenon, but this is lost as these results are presented. The authors might consider presenting the results separately for veterans and non-veterans, or providing a table of themes, with a column of illustrative quotations for each group and then comparing them afterwards, rather than switching back and forth within the text. • The authors need to be careful also of the language used throughout the results section, which implies that the differences were quantified and that differences between non-veterans and veterans were measurable. • Finally, the value of the caregiver data for this paper is unclear, and seems to add more confusion, rather than help corroborate the experiences. The caregiver experiences might best be presented as a separate paper, as I am sure this data is rich on its own. • Since veterans were divided into those with and without mental health problems, were any differences in the themes noted in these groups? (This speaks further to the justification for why these were separated.) Table 1:  • Are the gender data for the close companions column correct? If most participants are male, and most of the close companions are spouses, 85% male for the companions seems high? • Was data collected on veterans' civilian occupation after leaving the service? Did participants discuss any impact of this on their health? • Providing the total n's for each of the groups in the columns would be helpful Minor points:  • There are several small grammatical errors throughout the abstract and manuscript which need to be addressed.  o Ex., Abstract, "Setting"- there is an extra "in" in this sentence
--	--

REVIEWER	Kara Rudolph University of California, Davis, United States
REVIEW RETURNED	28-Jan-2019

GENERAL COMMENTS	These comments only pertain to the quantitative components of the manuscript.  - SF 12 data are missing for 18% of the sample. This limitation should be noted in the discussion. - Table 1. Please round to the nearest whole number for percentages. Otherwise, it gives the impression that have more data than actually have. - No information is given about the results of the statistical tests conducted. In the Methods Section, the authors write that they conduct t-tests, Fisher's exact tests, and Kruskal-Wallis tests. In the Results section, where Table 1 is being discussed, please give the name of the test, test statistic, and inference for each comparison being made.
--

VERSION 1 – AUTHOR RESPONSE

Referee 1

1. Recommendation that we include a justification for using a qualitative approach and how the sample size was determined

Thank you for this review point. We have now justified the use of a qualitative approach, to allow for the exploration of issues most salient to participants, and our sample size on pages 6 and 7.

2. Request for additional information about the recruitment of veterans with and without mental health problems.

It has been well established in previous studies that individuals with mental health problems often have comorbid physical health problems. The inclusion of veterans with mental health problems was actively sought in order to investigate whether older veterans with mental health problems also had poorer physical health and if this was believed to be linked to their occupation. We have made this point clearer on page 7. We acknowledge as a limitation in the discussion that the inclusion of non-veterans with mental health problems would also have been useful in clarifying this issue (page 16).

3. Query regarding how thematic saturation was determined

We describe the topic of thematic saturation on page 7 and describe how this was achieved. For clarity, saturation was determined when researchers, in the reviewing of successive data, did not observe any additional themes emerging.

4. Query regarding the context of the larger study

Due to word limit restrictions, we are not able to describe the larger scale project in detail. However, we have included a reference to the full study report should readers be interested. We anticipate that over time we will produce a number of papers from the larger study but considered this one as being particularly of interest and thus we set out to publish this first.

5. Query regarding the clinical context that participants were recruited from and the criteria used to identify potentially eligible participants by the clinical care team

We describe on page 8 that participants were recruited from National Health Service (NHS) GP surgeries, walk in centres, community groups or mental health services. We detail that the clinical care team identified potentially eligible participants using study inclusion/exclusion criteria (page 8)

6. Suggestion that additional information regarding the process of thematic analysis is included

Thank you for this request. We have now listed the steps undertaken throughout the thematic analysis process, the primary researcher who proposed initial codes and themes, and how discussions about the findings with co-authors were incorporated as part of the peer debriefing process on pages 8 & 9.

7. Suggestion that we reorganise the presentation of participant quotes to improve clarity for readers.

We thank the reviewer for this feedback and have revised the results section with all participant quotes now listed in Table 2. We have ensured that additional non-veteran quotes are listed. We feel

these changes have significantly improved the clarity of the results section for readers, with comparisons between veterans and non-veterans more clearly portrayed. We hope that the reviewers are satisfied with our response but are happy to attempt to clarify further if that is felt to be necessary.

8. Recommendation that we revise our language throughout to remove references to quantified differences

We have tempered our language throughout the results section to more clearly describe the observed thematic differences in experiences between veterans and non-veterans.

9. Query whether the experiences of close companions detracts from the aim of the study to explore the impact of occupation on wellbeing later in life.

We have given this point a great deal of consideration. Having considered our findings, we believe that the inclusion of close companions' data serves to enrich and triangulate the views of veterans/non-veterans about the diverse impact of military and non-military occupations on physical health. Data from close companions also illustrates the widespread effects that physical health problems can have not only for the employee themselves, but on their family, and how these effects manifest in later years. We have amended the manuscript to more effectively portray the views of close companions and hope that our changes better delineate their experiences. Our department has done quite a bit of work on the impact of military service on the families of veterans, and those who are still serving, and consider this to be a topical and important subject.

10. Query whether there were differences between veterans with and without mental health diagnoses.

Few thematic differences were found between veterans with and without mental health problems. To more clearly detail this, we have added a statement on page 12.

11. Suggestion regarding the presentation of data in Table 1

We have amended an editorial error to reflect that 85% of close companions were female. All veteran participants were no longer serving in the Armed Forces and those who had civilian occupations worked in these professions post-service. For the sake of clarity, we have provided the total sample size of each subsample in the heading column of Table 1.

12. Recommendation that minor editorial lapses are addressed

We have amended editorial lapses throughout the manuscript and thank the reviewer for noting the presence of these.

Referee 2

1. Suggestion that missing data is discussed as a limitation in the discussion
Thank you for this review point; we have noted this as a limitation in the discussion on page 18.

2. Suggestion that nearest whole numbers are used for percentages in Table 1.
We have amended Table 1 to reflect whole numbers where percentages are given.

3. Recommendation that we provide the names of the statistical tests carried out in the Results section.

We have clarified for readers that t-tests were used to compare scores on the SF-12 and EQ5D-VAS. We have also, for the sake of clarity, provided the test statistics in Table 1.